# Safety culture in French nursing homes: A randomised controlled study to evaluate the effectiveness of a risk management intervention associated with care

Delphine Teigné[1,2], Guillaume Mabileau[1], Marion Lucas[1], Leila Moret[1,2,3], Noémie Terrien[1] *

1 QualiREL Santé, Saint Jacques Hospital, Nantes, France, 2 Public Health Department, University Hospital of Nantes, Nantes, France, 3 UMR INSERM U1246-SPHERE "methodS for Patients-centered outcomes & HEalth REsearch", University of Nantes, University of Tours, Nantes, France

* nterrien@qualirelsante.com

## Abstract

### Background

French Nursing Homes (NHs) are in the early stages of implementing their Risk Management (RM) approach. A regional structure, which was mandated to provide independent support in RM, designed a training package.

### Objective

To study the impact of the RM training package on safety culture (SC) in NHs and drivers for improvement in SC scores.

### Method and analysis

This randomised controlled study targeted French NHs. Inclusion criteria were voluntary participation, no external support provided on the topic of adverse incidents upstream of the project, and the commitment of top management to its implementation. The 61 NHs were randomly allocated to one of two groups: the first benefited from a training package; support was given to the second after the impact measurement. Seven dimensions of SC were measured, at an 18-month interval, using the validated Nursing Home Survey on Patient Safety Culture questionnaire (22 items), which was administered to all of the professionals working in NHs. Eleven variables were captured, relating to the structural profile of the NH, the choices of top management in terms of healthcare safety, and the implementation of the system. Further modelling identified predictive factors for changes in SC scores.

### Results

95% of NHs completed both rounds of the questionnaire. The dimension *Feedback and communication about incidents* (SC = 85.4% before the intervention) significantly improved (+2.8%; *p* = 0.044). Improvement in the dimension *Overall perceptions of resident safety*–

**Data Availability Statement:** All relevant data are within the paper and its Supporting information files.

**Funding:** This study is part of the EPHAGE French research project on care system performance. The project was coordinated by University Hospital of Nantes, and funded by the Direction Général de l'Offre des Soins (DGOS; https://solidarites-sante. gouv.fr/ministere/organisation/organisation-des-directions-et-services/article/organisation-de-la-direction-generale-de-l-offre-de-soins-dgos) over the period 2015–2017. The funders had no role in study design, data collection and analysis, decision to publish, or preparation of the manuscript.

**Competing interests:** The authors have declared that no competing interests exist.

**Abbreviations:** AEAC, Adverse events associated with care; NH, Nursing Home; NHSOPS, Nursing Home Survey on Patient Safety Culture; RM, Risk Management; RSS, Regional Support Structures; SC, Safety Culture.

*organizational learning* was close to significant (+3.1%; *p* = 0.075). Drivers for improvement in scores were a pre-existing quality improvement approach, and a steering group that showed RM leadership.

## Conclusions

The system appears to have improved several dimensions of SC. Our findings are all the more important given the current crisis in the healthcare sector.

## Trial registration

Retrospectively registered as NCT02908373 (September 21, 2016).

## Introduction

Patient harm due to unsafe care is a growing public health problem [1]. The World Health Organization has established its Global Patient Safety Action Plan 2021–2030 for health care settings (primary care, nursing homes, hospitals) according to 7 guiding principles. Instilling a safety culture (SC) in the design and delivery of healthcare is one of these 7 guiding principles [1]. SC is a multidimensional concept, and there is currently no consensus on the definition, number, nature and denomination of its dimensions [2]. The European Society for Quality in Health Care defines it as, "an integrated pattern of individual and organisational behaviour, based upon shared beliefs and values that continuously seeks to minimise patient harm, which may result from the processes of care delivery" [3]. Leadership commitment, transparency, and learning from Adverse Events Associated with Care (AEAC) are essential components of a SC [1]. In France, the National Patient Safety Programme 2013–2017 focused on the fight against the advent of AEACs *via* the Risk Management (RM) approach [4].

Faced with an ageing world population and an increase in chronic disease, health systems are constantly evolving [5]. In France, the accommodation provided to dependent elderly people who require day-to-day care has undergone many changes. French nursing homes (NHs) accommodate elderly people who have lost their physical and/ or mental independence, and who can no longer live in their own home [6]. The literature highlights that there is a long way to go before they implement the RM approach [7]. Furthermore, it emphasises the importance of adapting the AEAC concept to French NHs, and to the specific context of care and assistance provided on a daily basis in the place of residence [6].

The assessment of SC helps to raise awareness of patient safety issues, and identify potential areas for improvement. According to Shortell *et al.*, it is one of the key pillars for achieving quality and RM objectives [8]. In this regard, various tools have been developed to measure it. In particular, the American Agency for Healthcare Research and quality has developed a questionnaire tailored to NHs: the Nursing Home Survey on Patient Safety Culture (NHSOPS) [9]. Its adaptation to different contexts has been the subject of various articles [10–12], including a study in France [13]. The psychometrically-validated French version (NHSOPS-F) includes 22 items that measure seven dimensions of SC [13].

In 2016, the French government created several Regional Support Structures (RSS) spread throughout the country. The main role of these structures is to support healthcare professionals in hospitals and medico-social establishments (including NHs) by providing expertise on quality improvement and patient care safety [14]. Only a few examples of support systems for healthcare professionals have been reported in the literature, and they mainly concern team

training, the creation of patient safety teams, or patient safety education programs implemented in hospitals—and even fewer have been evaluated for their effectiveness [15]. Moreover, few target NHs [16] or care at home [17].

As, to the best of our knowledge, there are no reports of schemes that propose a comprehensive RM structure tailored to NHs, QualiREL Santé (a RSS dedicated to the quality and safety of care) designed a training package to support French NHs in the implementation of their RM system. The scheme aimed to provide a long-term strategy for structuring and reporting AEAC of different levels of severity, and was based on the four axes found in the Shortell model [8]. It targeted staff at all hierarchical levels, whatever their role in supporting the elderly person. The aim was to develop SC among professionals and, more specifically, improve overall perceptions of resident safety. It was therefore important to study the impact of the implementation of the intervention on SC.

Any recommendations for improvements to the support system, or its successful implementation, must be underpinned by an understanding of the parameters that influence the final results [18–20]. These parameters may be related to the structural profile of the NH, its appropriation of the system, or the strategic healthcare safety choices made by top management [21].

The main objective of our research was, therefore, to study the impact of the implementation of an RM support system, which was deployed as training package, on the SC of professionals. We measured the evolution of SC in a group of NHs supported in their RM strategy; and the relation between the change in SC scores and the characteristics of the NH. Our results provide an insight into the actions and strategies that could be adopted to develop SC in NHs.

## Method

### Participating nursing homes

A total of 61 NHs from two French administrative divisions (*départements*) (Loire-Atlantique and Vendée) participated. Inclusion criteria were: i) voluntary participation in September 2015 to benefit from a system to support the implementation of a RM approach, ii) a written commitment by the NH's top management to implement a RM approach; and iii) no external support had been provided by QualiREL Santé on the topic of adverse incidents upstream of the project. The characteristics of French NHs have been described in earlier work [21], and their legal status can vary (independent, part of a group, attached to a public hospital). They also differ in terms of the number of beds, the number of staff, and the ratio of staff to residents. High-dependency residents are scored according to their degree of dependency [22].

### Tools

**The training package.**   The overall objective of the training package was to outline a strategy to structure the reporting and analysis of AEAC in the long term. In particular, the goals were to: i) provide a definition of AEAC tailored to the context of French NHs, with illustrative examples [6]; ii) to propose scales for the severity and frequency of AEAC [23]; iii) to provide formal tools for a reporting and analysis system; and, iv) to create a steering group of NH professionals that would be responsible for the analysis of AEACs and the definition of an action plan.

The training package was expected to lead to improvements in SC among NH staff, particularly overall perceptions of resident safety. It consisted of 20 hours (for each NH) of methodological support provided by two facilitators from the research team. Support was structured into four interventions. The initial session [1] raised awareness among top management of the importance of taking ownership of the project. The aim of the second session [2] was to raise

Table 1. Presentation of the training package categorized according to the 4 axes in Shortell's model.

| Axes in Shortell's model [8] | Support provided by QualiREL Santé | Targeted themes | Tools provided to the NH |
|---|---|---|---|
| Strategic | Session 1 | • Raise the awareness of top managers of the importance of the project's implementation (challenges are included in the establishment's policy, definition of objectives in the Nursing Home's quality-risk Program). | • Project implementation commitment charter signed by top management. |
| Cultural | Sessions 2 and 3 | • Raising awareness of the value of reporting.<br>• Raising awareness of a positive culture of error.<br>• Raising awareness of the importance of feedback. | • Sample charter to encourage AEAC reporting.<br>• Communication support, including feedback. |
| Structural | Session 1 | • Identification of a peer as a contact point for the project | |
| | Session 3 | • Setting up a RM steering group. | • Template operating charter for the RM steering group.<br>• Template procedure for reporting AEACs.<br>• Template AEAC report form. |
| Technical | Session 3 | • Train members of the RM steering group in the monitoring and analysis of AEACs. | • Computerized database for recording and monitoring AEACs. |
| | Session 4 | • Provide support during the first analysis. | |

AEAC: adverse events associated with care RM: Risk Management

awareness among all NH staff of the need to report AEAC, and identify volunteers who would join the steering group. The third session [3] was run for members of the RM steering group; here the aim was to train them in how to structure their work and analyse AEAC. Then, the aim of the final session (4) was to observe how the steering group handled a critical AEAC that had occurred in the NH, and provide support in its analysis. At the end of each intervention, dedicated tools that had been specifically designed for the package were presented, and made available to participants. Table 1 presents the training package as a function of the four axes in Shortell's model [8]. The four sessions were implemented over a period of nine months.

Our recommendations for the implementation of the training package were as follows: as many NH staff (including top management) as possible should attend the awareness-raising session; the RM steering group should be composed of different staff categories (nurses, local doctors, care assistants, staff responsible for catering and laundry, secretaries, etc.); and the steering group should have met at least once (and, if possible, should have run an analysis of an AEAC that had occurred in the NH) before taking part in the fourth session.

**The Nursing Home Survey on Patient Safety Culture.** The French version of the psycho-metrically-validated NHSOPS includes 22 items that measure the following seven dimensions of SC [13]. Dimension 1 (*Overall perceptions of resident safety and organizational learning*) explores the ability of the facility to take account of its mistakes and make changes, and the overall quality of care provided for residents. Dimension 2 (*Handoffs*) assesses communication for resident care both within the NH, and with external structures. Dimension 3 (*Teamwork*) explores support, mutual respect and collaboration among colleagues. Dimension 4 (*Supervisor expectations and actions promoting resident safety*) captures supervisors' acknowledgement of compliance with procedures, and their openness to ideas for improvement. Dimension 5 (*Compliance with procedures*) explores general compliance with mandatory procedures implemented in the facility, regardless of any operational difficulties encountered by professionals. Dimension 6 (*Staffing*) captures the number of staff in relation to the workload. Finally,

Dimension 7 (*Feedback and communication about incidents*) explores how potential or actual incidents are dealt with, and solutions are shared.

This self-administered questionnaire is aimed at professionals working in French NHs (both staff employed directly by the NH, and external contracted professionals). It is divided into four sections: "Working in your NH", "Communication", "Your hierarchy" and "Your NH". There are 22 items, which are scored on a 5-point Likert scale (*never, rarely, sometimes, most of the time, always*–or–*do not agree at all, do not agree, agree up to a point, agree, completely agree*). "Does not apply" and "Don't know" are other response categories. Finally, the questionnaire includes items relating to the sociodemographic characteristics of respondents.

## Study design

The study followed the randomised controlled study design. It ran from September 2015 to November 2017.

**Randomization.**   The 61 NHs were randomly allocated into two groups; the first group received support in 2016 (experimental group– 31 NHs), and the second group received support after the impact assessment in November 2017 (control group– 30 NHs). The homogeneity of the two groups in terms of distribution by *département* and legal status was checked.

**Implementation of the training package.**   The operational implementation of the support system ran over a period of nine months. Two facilitators from the research team spent time in each NH. The latter had a training and coaching role in the implementation of the support system (from sessions 1 to 4). Two professionals were appointed by the NH to be the main point of contact during the project.

**SC measurement.**   The same NHSOPS-F questionnaire was administered to the two groups prior to the implementation of the training package (January 2016), and again around 18 months later (September 2017). It was important that the NH took ownership of the support system over time. The methodology for the distribution and collection of questionnaires has been extensively described in an earlier article [21]. To summarize, in each NH the two contact points were responsible for: distributing the questionnaire to each professional working in the NH; promoting and following-up the survey; centralizing the completed questionnaires; and returning them to the research team [21]. The survey was conducted over a six-week period. Each completed questionnaire was given a number to ensure anonymity. Responses were recorded by an external technician, and data capture was blind, in duplicate, with cross-checking. Because SC was measured at the NH level, the analysis was performed by compiling all of the responses from professionals working in the same NH, with the aim of comparing responses from the same NH between the two measures.

## Descriptive variables for NHs in the experimental group

Eleven descriptive variables were collected at the beginning of the project in 2016. Five related to the structure of the NH: *bed capacity*; *legal status*; *being part of a group or not*; the *mean dependency score*; and *the staff/ resident ratio*. Six other variables documented top managers' strategic choices in terms of safety of care: *an established policy of ongoing improvement in quality and RM*; *an active quality improvement approach*; and *an active RM approach*, approaches facilitated by the presence of either: *an appointed RM quality officer* (a member of staff with no specialized RM quality qualifications), *a RM quality specialist* (a graduate with RM quality qualifications), or *an external quality and RM service provider*. The categorization of these variables is presented in Table 2.

**Table 2. Descriptive variables of nursing homes and the implementation of the scheme.**

| Descriptive variables in 2016 | Case group n = 28 NHs |
|---|---|
| | n (%) or mean (SD) |
| ***Beds**\* | |
| ≥80 | 15 (53.5%) |
| <80 | 13 (46.5%) |
| ***Legal status*** \* | |
| Private or public, independent or regional | 21 (75%) |
| Attached to a public hospital | 7 (25%) |
| ***Part of a group*** \* | |
| No | 13 (46.5%) |
| Yes | 8 (53.5%) |
| Hospital-based | 7 (25%) |
| ***Dependency score**\* | |
| Mean (SD) | 625 (75.54%) |
| NA | 1 (3.6%) |
| ***Staff/resident ratio*** \* | |
| Mean (SD) | 0.74 (0.19%) |
| NA | 0 (0.0%) |
| ***Unqualified RM/quality officer*** \*\* | |
| No | 7 (25%) |
| Yes | 21 (75%) |
| NA | 0 (0%) |
| ***Qualified RM/quality specialist*** \*\* | |
| No | 16 (57.1%) |
| Yes | 12 (42.9%) |
| NA | 0 (0%) |
| ***External RM/quality service provider*** \*\* | |
| No | 28 (100%) |
| Yes | 0 (0%) |
| NA | 0 (0%) |
| ***Established policy of ongoing improvement in quality and RM*** \*\* | |
| No | 9 (32.1%) |
| Yes | 19 (67.9%) |
| NA | 0 (0%) |
| ***Active quality improvement approach*** \*\* | |
| No | 4 (14.3%) |
| Yes | 24 (85.7%) |
| NA | 0 (0%) |
| ***Active RM policy*** \*\* | |
| No | 11 (39.3%) |
| Yes | 17 (60.7%) |
| NA | 0 (0%) |
| **Change in descriptive variables for nursing homes in the experimental group in 2017** | **Case group n = 28 n (%) or mean (SD)** |
| ***Change in the staff/ resident ratio**\* | |
| Mean (SD) | +0.03 (0.08) |
| NA | 4 (14.3%) |
| ***Change in the dependency score**\* | |

*(Continued)*

**Table 2.** (Continued)

| | |
|---|---|
| Mean (SD) | +14.3 (35.7) |
| NA | 5 (17.9%) |
| **Variables related to the implementation of the system in the experimental group in 2017** | **Case group n = 28 n (%) or mean (SD)** |
| *% staff present at the awareness-raising session (session 2)***\*\*** | |
| Mean (SD) | 49.5 (22.5) |
| NA | 0 (0.0%) |
| *% top management present at the awareness-raising session (session 2)***\*\*** | |
| No | 11(39.3%) |
| Yes | 17 (60.7%) |
| *% staff wanting to use the knowledge they have gained***\*\*** | |
| Mean (SD) | 91.1 (12.6) |
| NA | 2 (7.1%) |
| *% staff showing leadership in the RM approach***\*\*** | |
| Mean (SD) | 34.0 (24.1) |
| NA | 2 (7.1%) |
| *Mature RM steering group***\*\*** | |
| No | 10 (35.7%) |
| Yes | 16 (57.2%) |
| NA | 2 (7.1%) |
| *Turnover among RM steering group members***\*\*** | |
| No | 21 (75%) |
| Yes | 5 (17.9%) |
| NA | 2 (7.1%) |
| *Unexpected disruptive events***\*** | |
| No | 15 (53.6%) |
| Yes | 8 (28.5%) |
| NA | 5 (17.9%) |

RM: Risk Management NA: no answer SD: Standard Deviation

\* Variables related to the structural profile of NHs

\*\* Variables related to strategic safety of care choices

## Change in descriptive variables for NHs in the experimental group in 2017

In 2017, change in scores for two variables related to the structural profile of the NH were captured: *the staff/ resident ratio*, and the *mean dependency score* (Table 2).

## Change in variables related to the implementation of the training package in the experimental group in 2017

As the project progressed, the following seven variables related to recommendations for implementing the system were measured.

1. The percentage of staff who attended the awareness-raising session.

2. The presence of top management at the awareness-raising session.

3. An anonymous questionnaire was distributed to each member of the steering group at the end of session 3. This questionnaire identified the percentage of professionals who "wanted to use the knowledge they had gained" or

4. who "felt able to pass on their knowledge to colleagues".

5. The degree of maturity of the steering group in implementing the approach was recorded by the two facilitators. This measure corresponded to the use of the systemic AEAC analysis tool that was made available between the 3rd and 4th sessions.

6. Facilitators also recorded the stability of the steering group, notably the withdrawal or replacement of its members.

7. Finally, the research team noted any contextual elements that could undermine the implementation of the system (for example, the relocation of the NH, turnover among top management, absenteeism among local management, and a lack of commitment from top management).

The categorization of these seven variables is shown in Table 2. The research team developed bespoke tools to collect this data, and these tools are not licensed.

### Data analysis

The calculation of SC scores has been described in detail in previous work [21]. All analyses were performed at the NH level. Following the method given in the original American version of the NHSOPS [9], scores were calculated as the average percentage of positive responses on all items in each dimension.

The validity of the model constructed from data collected from the first round of questionnaires [13] was rechecked against data from the second round. Four statistical indices were taken into account in order to verify the fit of the model to the data: Steiger's Root Mean Square Error of Approximation (RMSEA), and the Standardized Root Mean Squared Residual (SRMR): fit was considered good when these values were <0.1, and very good when <0.05. The Tucker Lewis Index (TLI) and the Comparative Fit Index (CFI) were considered an acceptable fit when >0.9 [24]. Cronbach's α coefficients were calculated to assess internal consistency [25]. Consistency was considered acceptable when $\alpha > 0.70$ [26]. SC scores were calculated for each NH that completed the two questionnaires in 2016 (baseline) and in 2017 (following the intervention). Scores for the seven dimensions for each NH were compiled at the group level (experimental group = 30 NHs; control group = 28 NHs). Thresholds to discriminate the levels of development of the dimensions were set at: SC <50% (underdeveloped); $50\% \leq SC < 75\%$ (developing); and $SC \geq 75\%$ (developed) (13).

Changes in scores for the seven dimensions were evaluated using a paired Student's *t*-test. A threshold of 0.05 was used to establish a significant difference between the two measures.

Descriptive statistics were run for all variables.

Linear regression models were used to assess variables associated with each change in the SC score in univariate analyses to identify potential predictors at a significance level of 0.2. Multivariate analyses were then performed according to the backward stepwise method, in order to identify models that best fitted the data, and keep the most relevant factors associated with each change in the SC score, at a significance level of 0.05. Validity and goodness of fit were checked using adjusted $R^2$ and an F-test.

All statistical analyses were performed using R software (version 4.1.1).

### Ethics approval and consent to participate

The EHPAGE study is registered with the Cnil (Commission Nationale de l'Informatique et des Libertés) under ref. 915719 (December 1, 2015). Our research protocol was approved by the Advisory Committee on Health Research Information Processing, and an ethics committee

(*Groupe Nantais d'Ethique dans le Domaine de la Santé* [University Hospital of Nantes]). Professionals were informed about the purpose of the research in advance. Participation was voluntary and informed consent was implied upon completion of the questionnaire (a procedure approved by the ethics committee). According to articles L1121-1 and R1121-2 of the French Public Health Code, Institutional Review Board approval was not necessary.

## Results

### NH participation

**Participation in the implementation of the training package.** By November 2017, 30 of the 31 NHs in the intervention group had benefited from the entire support package (Fig 1).

**Participation in the NHSOPS-F questionnaire.** Out of the 61 NHs operating in 2015, 58 (95%) completed the two rounds of questionnaires (Fig 1).

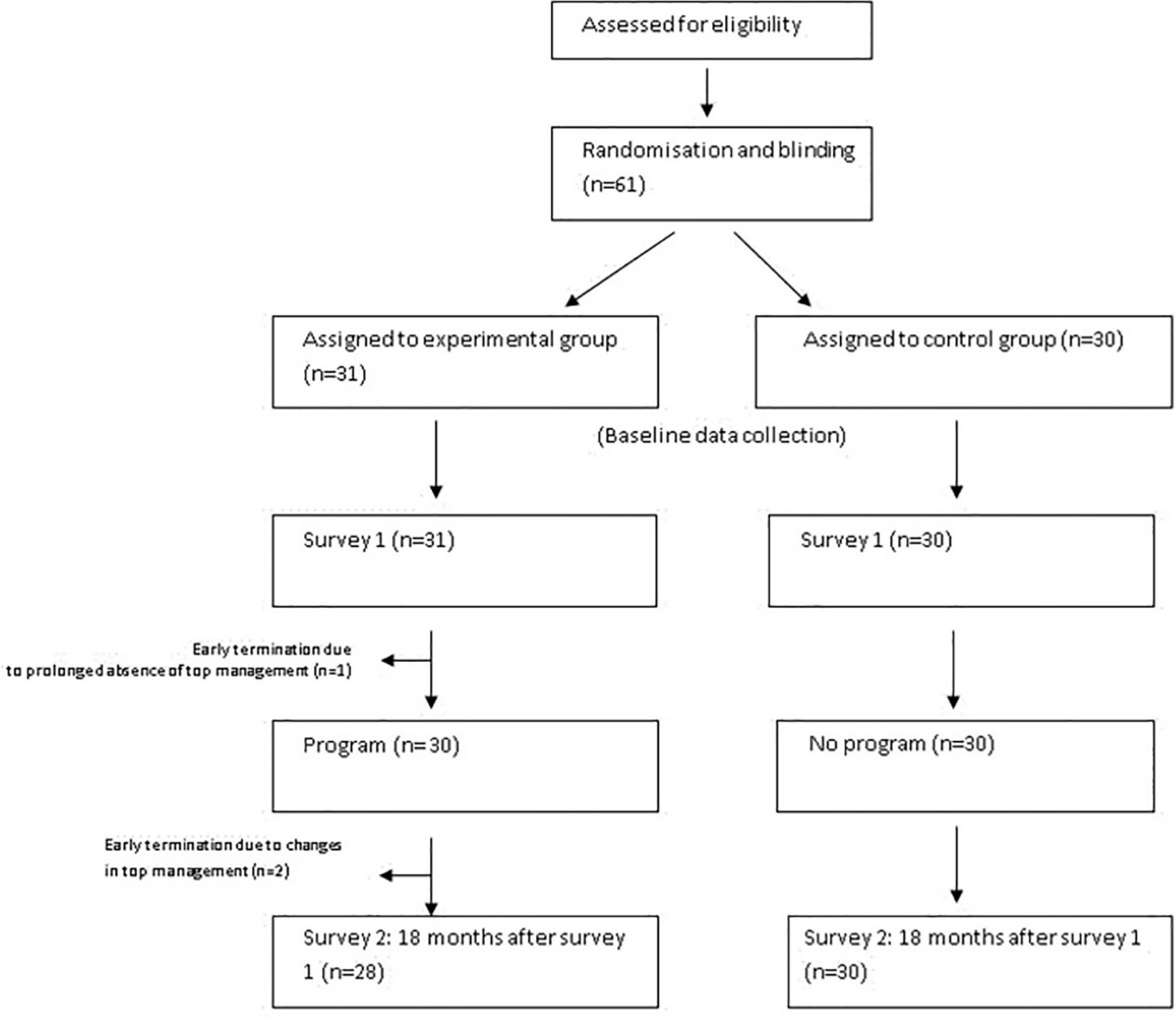

**Fig 1. Flow chart of the study in control and intervention groups.**

The questionnaire was administered to 3,390 professionals in 2016 (1,946 responded) and 3,140 professionals in 2017 (1,621 responded). Of those who responded in 2016, 1,012 were in the experimental group and 934 were in the control group. In 2017, these figures were 754 (experimental group) and 867 (control group). The median overall participation rate for all NHs was 64% in 2016 and 56.5% in 2017. S1 Table compares the characteristics of respondents in 2016 and 2017. The main differences between 2016 and 2017 concern: the percentage of paramedical professionals (+3.6%; $p = 0.038$) and doctors (−1.1%; $p = 0.001$); the percentage of professionals aged under 25 (+2.3%; $p = 0.002$) and between 36 and 45 (−3.1%; $p = 0.026$); and professionals working 15 hours or less per week (−2%; $p = 0.007$). All other categories were not statistically different at the 5% alpha level.

## Descriptive variables

**Descriptive variables for NHs in the experimental group.** Of the 28 NHs in the experimental group that completed the two questionnaires, seven were hospital-based, and 21 had another status. Bed capacity was equal to or over 80 for 15 NHs. The mean staff/ resident ratio was 0.74 in 2016 and 0.77 in 2017. The mean dependency score was 625 in 2016 and 639.3 in 2017. In 2016, an active Quality improvement approach was in place in 24 NHs, and there was an active RM approach in 17. For 12 NHs, these policies were facilitated by a specialized RM Quality graduate, while in 21 NHs they were the responsibility of a salaried NH professional with no specific RM Quality qualifications. These results are presented in Table 2.

**Change in descriptive variables for NHs in the experimental group.** The analysis found that in the 28 NHs in the experimental group, there was no significant change in the staff/ resident ratio, or the mean dependency score during the research period (Table 2).

**Variables related to the implementation of the training package.** On average, 49.5% of NH professionals attended the awareness-raising session; top management were present in 60.7% of NHs. On average, 7.2 professionals were involved in the steering group, and on average, 96.2% of steering group members completed the questionnaire that was administered after session 3. The analysis found that 91.1% of steering group members wanted to put into practice the knowledge they had acquired during the training sessions, and 34% showed leadership in the RM process. Finally, following session 4, the steering group was considered mature in 57.2% of NHs, and stable in 75%. Table 2 documents the seven variables used to measure the impact of the intervention.

## SC scores, group homogeneity

The validity of the model constructed in 2016 was confirmed by data from 2017 (S2 Table).

Table 3 presents scores for the seven SC dimensions for experimental and control groups in 2016 and 2017. The analysis found no statistically significant differences between the two groups at the beginning of the project.

Fig 2 reports changes in SC scores between the administration of the two questionnaires in experimental and control groups. There were no significant changes in the control group between the two runs. However, in the experimental group, scores for Dimension 7 (*Feedback and communication about incidents*; $p = 0.044$) did increase significantly, indicating an improvement. Scores for two other dimensions (Dimensions 1 and 4; *Overall perceptions of resident safety—organizational learning* and *Supervisor expectations and actions promoting resident safety*) also increased, and were close to significance ($p = 0.075$ and $0.067$) (Fig 2). Finally, scores for Dimension 2 (*Handoffs*) fell, and the fall was close to significant ($p = 0.089$).

**Table 3. Scores for each of the seven SC dimensions by group and survey round.**

| Group (number of NH) | Survey number (date) | SC scores by dimension (%) | | | | | | |
|---|---|---|---|---|---|---|---|---|
| | | Dimension 1 | Dimension 2 | Dimension 3 | Dimension 4 | Dimension 5 | Dimension 6 | Dimension 7 |
| | | *Overall perceptions of resident safety— organizational learning* | *Handoffs* | *Teamwork* | *Supervisor expectations and actions promoting resident safety* | *Compliance with procedures* | *Staffing* | *Feedback and communication about incidents* |
| | | mean Q1 Q3 | mean Q1 Q3 | mean Q1 Q3 | mean Q1 Q3 | mean Q1 Q3 | mean Q1 Q3 | mean Q1 Q3 |
| Total (n = 58 NHs) | 1 (January 2016) | 75.8% [69.5%; 87.8%] | 76.9% [72.0%;84.2%] | 60.4% [48.8%;72.4%] | 79.2% [72.7%;87.4%] | 55.0% [46.0%;62.9%] | 11.8% [3.7%;17.7%] | 84.8% [79.3%;91.9%] |
| Experimental (n = 28 NHs) | 1 (January 2016) | 75.5% [70.8%;90.3%] | 78.7% [74.0%;85.9%] | 60.9% [48.8%;70.5%] | 77.6% [72.7%;88.8%] | 56.1% [47.8%;64.1%] | 11.6% [4.2%;18.4%] | 85.4% [80.3%;94.6] |
| | 2 (September 2017) | 78.6% [75.0%;88.7%] | 75.9% [72.1%;82.9%] | 58.1% [43.5%;74.0%] | 82.8% [76.4%;89.7%] | 54.1% [43.1%;64.9%] | 10.3% [3.0%;16.5%] | 88.2% [83.7%;93.7%] |
| Control (n = 30 NHs) | 1 (January 2016) | 76.0% [69.1%;86.6%] | 75.3% [70.7%;83.2%] | 59.9% [51.0%;72.4%] | 80.8% [77.9%;87.1%] | 53.9% [44.4%;59.5%] | 12.0% [3.6%;16.6%] | 84.2% [79.6%;91.3%] |
| | 2 (September 2017) | 72.3% [67.2%;86.0%] | 74.9% [70.3%;81.9%] | 57.9% [47.3%;65.2%] | 79.6% [76.5%;87.4%] | 51.7% [44.4%;60.7%] | 12.0% [3.4%;16.9%] | 85.1% [80.5%;92.3%] |

SC: Safety Culture NH: Nursing Home SC < 50%: underdeveloped 50% ≤ SC < 75%: developing SC ≥ 75%: developed

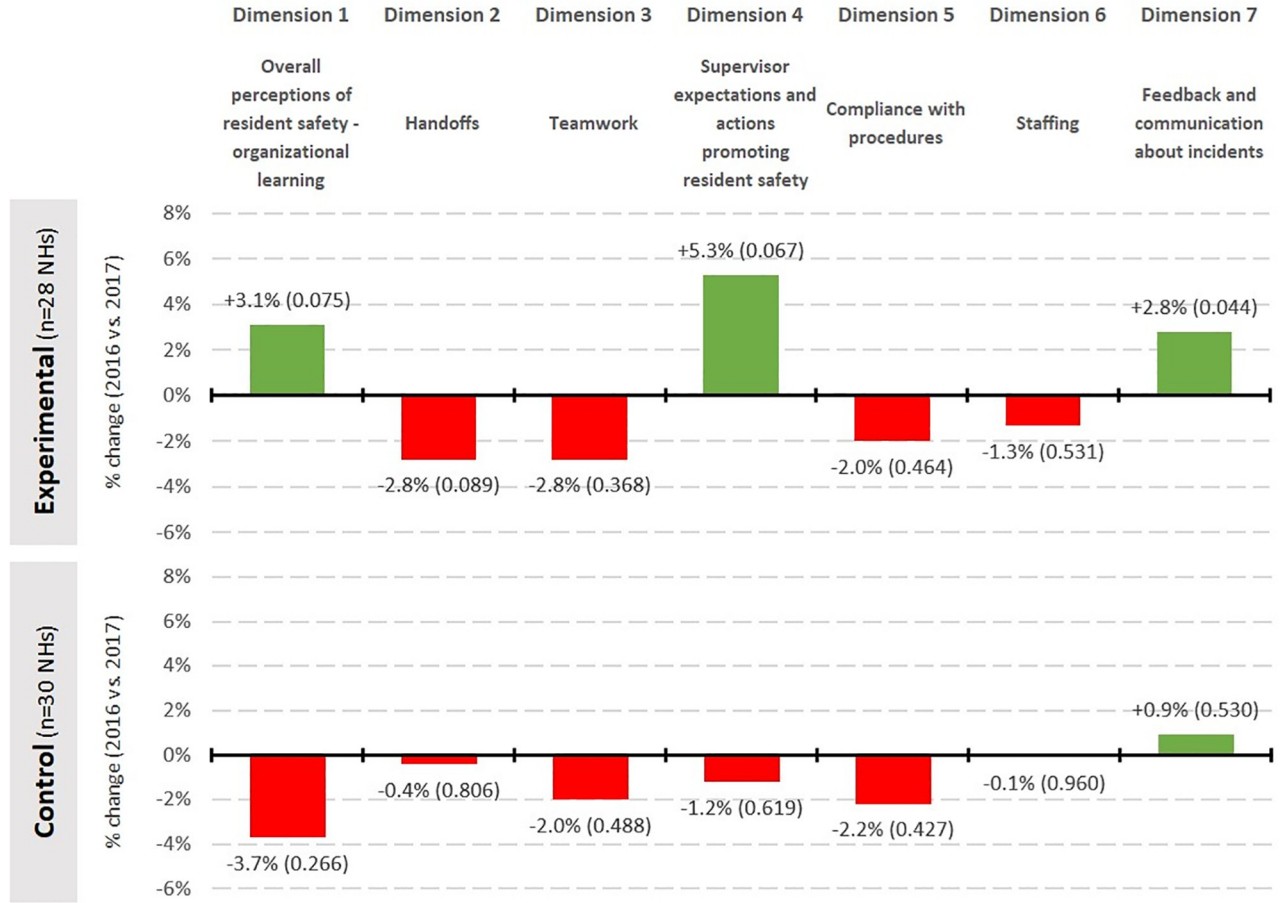

**Fig 2. Change in SC scores between 2016 and 2017 for each dimension and for each group.**

## Relationships between variables and change in SC scores

The results of the final multivariate models are presented in Table 4, and details of the univariate models used to construct them are presented in S3 Table. The analysis found that scores for Dimension 1 (*Overall perceptions of resident safety—organizational learning*) improved significantly as a function of the percentage of members of the steering group who showed leadership, and in NHs with an active Quality improvement approach (respectively, β-coefficient = 0.14/9.58; 95%CI = [0.01;0.28]/ [0.89;18.27]). Scores for Dimension 3 (*Teamwork*) improved in cases where the NH had an established policy of ongoing improvement in Quality and RM (β-coefficient = −12.42; 95%CI = [0.71; 24.12]). Scores for Dimension 4 (*Supervisor expectations and actions promoting resident safety*) improved when the NH was hospital-based–but not when it was independent, or when fewer members of the steering group stated that they wanted to use the knowledge they had gained from the training sessions (β-coefficient = 15.53/−0.49; 95%CI = [2.91;28.15]/[−0.87;−0.11]). Scores for Dimension 5 (*Compliance*

**Table 4. Parameters included in the multivariate models that explain change in scores for the seven dimensions of safety culture (n = 28 NHs).**

| Parameter | | Coef. [IC 95%] |
|---|---|---|
| **Dimension 1: *Overall perceptions of resident safety—organizational learning*** | | *Adjusted R² = 20.7%.* |
| Active quality improvement approach | No | ref. |
| | Yes | 9.58 [0.89;18.27]* |
| % staff showing leadership in the RM approach | | 0.14 [0.01;0.28]* |
| **Dimension 2: *Handoffs*** | | |
| N/A | | |
| **Dimension 3: *Teamwork*** | | *Adjusted R² = 16.4%.* |
| Established policy of ongoing improvement in quality and RM | No | ref. |
| | Yes | 12.42 [0.71;24.12]* |
| Mature RM steering group | No | ref. |
| | Yes | 9.84 [−2.2;21.87] |
| **Dimension 4: *Supervisor expectations and actions promoting resident safety*** | | *Adjusted R² = 19.4%.* |
| Membership of a group | No | ref. |
| | Yes | 5.35 [−4.74;15.43] |
| | Hospital-based | 15.53 [2.91;28.15]* |
| % staff wanting to use the knowledge acquired | | −0.49 [−0.87;−0.11]* |
| **Dimension 5: *Compliance with procedures*** | | *Adjusted R² = 21.2%.* |
| Unexpected disruptive events | Yes | ref. |
| | RAS | 11.1 [−0.41;22.61] |
| % staff showing leadership | No | 0.22 [−0.02;0.46] |
| **Dimension 6: *Staffing*** | | *Adjusted R² = 54.1%.* |
| Active quality improvement approach | No | ref. |
| | Yes | 12.02 [4.46;19.59]* |
| Change in dependency score | | 0.09 [0.01;0.18]* |
| % staff wanting to use the knowledge acquired | | 0.25 [0.03;0.47]* |
| **Dimension 7: *Feedback and communication about incidents*** | | |
| N/A | | |

*Significant at 5%

**Significant at <0.1%

N/A: not analysable RM: Risk Management

*with procedures*) improved when no unexpected disruptive events occurred (β-coefficient = 11.1; 95%CI = [−0.41;22.61]). Scores for Dimension 6 (*Staffing*) improved as a function of the percentage of members of the steering group who wanted to use the knowledge they had acquired, when there was an active Quality improvement approach, and when the dependency score increased (β-coefficient = 0.25/12.02/0.09; 95%CI = [0.03;0.47]/ [4.46;19.59]/ [0.01;0.18]). The model explained 54.1% of variability in the data for Dimension 6.

## Discussion

The main objective of this article was to study the impact of the implementation of an RM support system on SC among NH professionals. The results show that there is a difference in SC scores between 2016 and 2017 for NHs in the experimental group, while there is no difference for NHs in the control group. This suggests that the intervention had an impact on the SC of the professionals who participated.

### Change in SC scores

The intervention was designed to improve scores for Dimension 1 (*Overall perceptions of resident safety*) included in the original American version of the questionnaire [9]. The validation of the French version that was carried out during the research project led to the aggregation of this dimension with the dimension *Organizational learning*. The new Dimension 1 (*Overall perceptions of resident safety—organizational learning*), developed at the beginning of the project (75.8%), improved (by +3.1%) after the implementation of the support system. For the research team, the system clearly improved scores for the theme *Overall perceptions of resident safety*, however, the additional theme prevented the identification of a clear improvement in the new, aggregated dimension. Moreover, the literature underlines that organizational learning requires a process of profound change at different levels of the NH [15], and that its development can only be measured on the scale of many years [2].

Improvements in one other dimension deserve to be highlighted. Dimension 7 (*Feedback and communication about incidents*), which was already developed at the beginning of the project (85.4%), significantly improved (by +2.8%). This change was most probably brought about by, among other actions, the awareness-raising sessions, the tools that were made available, and empowering a group of professionals to analyse AEACs and provide feedback to colleagues on the action plan.

These results are very encouraging. We were only able to identify 3 interventions in NHs that have been evaluated with respect to their impact on SC [27–29]. None of these studies highlighted significant changes in any dimension. In the case of healthcare facilities, several interventions were measured for their impact using a questionnaire similar to the NHSOPS [18, 29–32]. In general, earlier work only highlights significant changes in a few dimensions [30, 31]. Scores for the dimensions *Feedback and communication of errors* and *Overall perceptions of safety* significantly improved in the studies by Ulibarrena *et al.* and Xie *et al.* [18, 31], respectively. However, only Xie *et al.* [18] noted a significant improvement in 7 dimensions following a SC training program and a questionnaire the mainly targeted nurses.

### Drivers to improve SC scores

Knowledge of the explanatory factors driving the change in scores is timely from the point of view of improving the system, and its large-scale deployment.

Quality policyemerged as a lever for the development of two dimensions (*Overall perceptions of resident safety—organizational learning*, and *Staffing*). It should be noted that the support system is based on tools, such as action plans and monitoring, that are usually used in

Quality approaches. Furthermore, the fact that NHs are already familiar with this type of tool supports the successful implementation of the system. Being hospital-based contributed to improved scores for Dimension 4 (*Supervisor expectations and actions promoting resident safety*). In France, managers can be shared between NHs and hospitals. In this case, NHs naturally benefit from a more mature attitude to patient safety. Finally, the mean dependency score emerged as a factor in the evolution of SC, in particular Dimension 6 (*Staffing*). It should be noted that in France, an increase in the dependency of residents increases the financial aid paid to the NH, allowing the recruitment of more care professionals.

With respect to the implementation of the system, greater leadership among the steering group was consistent with improved perceptions of SC, particularly with regard to Dimension 1 (*Overall perceptions of resident safety—organizational learning*) and Dimension 5 (*Compliance with procedures*).

The literature highlights the main explanatory factors for a SC [19, 28]. Two studies have examined the factors driving the evolution of scores: the first was conducted in Norwegian NHs [28], the second in a French healthcare institution [33]. Despite differences between the factors studied in these earlier works and the present study, past findings support our results. Our study shows that the successful implementation of the RM approach is conditional on the existence of a quality approach in the NH, and a steering committee that demonstrates leadership. These personnel have a mature understanding of SC [33], and appear to be the key to the success of the intervention, both in the short and the long term [34]. The organisational initiatives that they put in place, in collaboration with other professionals, are all-important [28]. The literature encourages top management to create and support such environments, which encourage staff to feel responsible for the SC of residents [28].

## Strengths and limitations of the EHPAGE study

This research shows that this multimodal RM intervention, which ran over many months, was accepted and implemented by the majority of participants. The reasons for dropping out were independent of the scheme (notably due to issues linked to staff absences or changes in top management). The study also shows that the intervention can improve dimensions of SC among professionals working in NHs. Our results are all the more remarkable given that the study was conducted at the NH level (rather than the unit level, in a small number (58) of NHs, with a short period of time (18 months) between the two SC measures. Running the study over a period of at least three to five years [15], and with a larger sample of NHs could confirm our results. It would also allow us to improve the robustness of the explanatory factors, and avoid certain measurement hazards.

The questionnaire response rate (64% in 2016 and 56.5% in 2017), although comparable with other studies (66% [10], 69% [35]), may also constitute a selection bias. Due to financial and organisational constraints, researcher bias could not be avoided in this study. Finally, the fact that participation was voluntary, and that only two French *départements* were represented, means that the results cannot be extrapolated to the entire country.

## The application of the EHPAGE project in the current health context

The EHPAGE intervention can be improved. Consideration could be given to the integration of residents' families, or their representatives, into the system. The failure to include these external actors in the process of improving healthcare safety and quality is all the more damaging as they have a different perspective, and often have an insight into more intimate aspects of residents' lives. In addition, health systems will evolve towards more home and outpatient care, requiring increased coordination between all actors. Therefore, and following Storm

*et al.* [29], further thought needs to be given to a collaborative analysis that crosses organizational boundaries between different actors and structures, and examines how care can best be provided. Finally, the challenges associated with the quality of the working life of professionals who provide care should not be forgotten [36, 37].

## Conclusion

French NHs are in the early stages of implementing their RM approach. A regional structure, which was mandated to provide independent support in RM, designed a training package. This system, which is all the more important given the current COVID-19 pandemic, should ultimately improve the care of residents by developing a SC among NH professionals. Our findings show that the system was well-received, and implemented by participating NHs. Its impact was measured quantitively *via* the NHSOPS-F questionnaire. The analysis found that the improvement in Dimension 7 (*Feedback and communication about incidents*) was statistically significant, and that improvement in Dimension 1 (*Overall perceptions of resident safety —organizational learning*) was close to significant (+3.1%; $p$ = 0.044). Our findings highlighted that the system was all the more effective in improving SC if the NH had already initiated a Quality approach. They also emphasize that it is important for top management to work with professionals who show leadership in the implementation of the RM approach.

To conclude, it is now up to each participating NH to keep the process alive. The system, which is now fully digitized, is currently being adapted to Belgian hospitals by a structure that is similar to QualiREL Santé.

## Supporting information

**S1 Table. Respondent characteristics for the 2016 and 2017 surveys.**
(DOC)

**S2 Table. Indicators to assess the global fit of the final structural equation model on 2017 data (R output).**
(DOC)

**S3 Table. Details of univariate and multivariate models explaining the evolution of safety culture scores for the 7 dimensions according to NH parameters and the implementation of the system (n = 28 NH).**
(DOCX)

## Acknowledgments

The authors would like to thank all 61 NHs who took part in the EHPAGE project. The authors would like to thank Elaine Seery at *Traduction Edition Scientifique* for translation and editing services.

## Author Contributions

**Conceptualization:** Leila Moret, Noémie Terrien.

**Data curation:** Delphine Teigné, Guillaume Mabileau, Noémie Terrien.

**Formal analysis:** Guillaume Mabileau, Noémie Terrien.

**Investigation:** Delphine Teigné.

**Methodology:** Guillaume Mabileau, Noémie Terrien.

**Project administration:** Delphine Teigné.

**Resources:** Delphine Teigné, Marion Lucas.

**Supervision:** Leila Moret, Noémie Terrien.

**Validation:** Guillaume Mabileau, Marion Lucas, Leila Moret, Noémie Terrien.

**Writing – original draft:** Delphine Teigné, Guillaume Mabileau, Marion Lucas, Noémie Terrien.

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
