## [Decision Letter · Decision Letter 0]

25 Jul 2022

PONE-D-22-13984Safety culture in French nursing homes: a randomised controlled study to evaluate the effectiveness of a risk management intervention associated with carePLOS ONE

Dear Dr. Terrien,

Thank you for submitting your manuscript to PLOS ONE. After careful consideration, we feel that it has merit but does not fully meet PLOS ONE’s publication criteria as it currently stands. Therefore, we invite you to submit a revised version of the manuscript that addresses the points raised during the review process.

In the first paragraph of the conclusion section, it is stated with reference to the manuscript. The conclusion of the manuscript should be based on the findings and discussion of the manuscript, not by referring to another reference. Therefore, it is suggested that if it is necessary to use the first paragraph of the conclusion section, the corresponding paragraph should be moved to another part of the manuscript (discussion section) and the conclusion section of the manuscript should be based only on the results and discussion conducted in this manuscript.It is necessary to check the entire manuscript for typographical errors.It is suggested that the authors check the references and include all the desired fields, including the doi. ==============================

We look forward to receiving your revised manuscript.

Kind regards,

Jahanpour Alipour, Ph.D.

Academic Editor

PLOS ONE

Journal Requirements:

   "The authors would like to thank the DGOS for its help. The authors would like to thank all 61 NHs who took part in the EHPAGE project. The authors would like to thank Elaine Seery at Traduction Edition Scientifique for translation and editing services. "

   "This study is part of the EPHAGE French research project on care system performance. The project was coordinated by QualiREL Santé, and funded by the Direction Général de l’Offre des Soins (DGOS ; https://solidarites-sante.gouv.fr/ministere/organisation/organisation-des-directions-et-services/article/organisation-de-la-direction-generale-de-l-offre-de-soins-dgos) over the period 2015–2017. The funders had no role in study design, data collection and analysis, decision to publish, or preparation of the manuscript."

Reviewers' comments:

Reviewer's Responses to Questions

**Comments to the Author**

1. Is the manuscript technically sound, and do the data support the conclusions?

Reviewer #1: Partly

Reviewer #2: Yes

2. Has the statistical analysis been performed appropriately and rigorously? 

Reviewer #1: Yes

Reviewer #2: No

3. Have the authors made all data underlying the findings in their manuscript fully available?

Reviewer #1: Yes

Reviewer #2: Yes

4. Is the manuscript presented in an intelligible fashion and written in standard English?

Reviewer #1: No

Reviewer #2: Yes

5. Review Comments to the Author

Reviewer #1: Dear authors,

The manuscript was fully reviewed. The scope of manuscript is interesting but it needs corrections, specially, discussion section needs correction. The review comments are in the manuscript. Best

Reviewer #2: Dear editors, thank you for giving me a chance to review this manuscript, here are my comment.

Reviewer Comments for a research entitled: Safety culture in French nursing homes: a randomized controlled study to evaluate the 2 effectiveness of a risk management intervention associated with care.

1. In the abstract section line number 30-31 how voluntary participant be inclusion criteria and this may affect the principle of random allocation(justify it)

2. In the result of abstract section of the line number 41-43, <<after intervention="" the="">3. In line number 53 to 66 in introduction section it is better to write about safety culture implementation in healthcare system and its importance in risk management rather than writing the characteristics of the population in that administrative region.

4. Is the tool is validated in your research context?

5. How determine the sample size for this study purpose?

6. Line number 124, randomization might be affected using voluntary participation of study participants as inclusion criteria rather than other exposure status and please explained it how was the selection process

7. Do you control researcher bias in assignments’ of in intervention in exposed group and unexposed group this is due to the researches should remain unaware of which treatment was given for which group until the study is completed?

8. Regarding to ethical concern of the research that involve human subject ethical concern is mandatory what types of consent you obtained from the study participants and before or after completion of the questioner participants was give their consent?

9. In line number 300-304, the result showed that presence of high non- response rate and did this affect your result?

10. In your discussion section it is better to start the discussion with the purpose of conducted this research followed by the main finding of your research.

In general the paper is very interesting and it’s very important to address problems of related with safety culture implementation in healthcare system (especially involvement of top management in SC promotion).

1. The article is needs rewrite in short.

2. The discussion part is shallow and it is better to add different literatures related with the concerned variables and it is so long but it needs short and precise to the concern.

3. Limitation and strength is so long it is better write in short .

Therefore, if this concern is address the paper will accepted for publication in your esteemed journal</after>

6. PLOS authors have the option to publish the peer review history of their article (what does this mean?). If published, this will include your full peer review and any attached files.

Reviewer #1: No

Reviewer #2: No

---

## [Author Response · Author response to Decision Letter 0]

28 Sep 2022

General comments

Below we set out our point-by-point responses to the reviewers’ feedback, we have the following general comments:

• In the first paragraph of the conclusion section, it is stated with reference to the manuscript. The conclusion of the manuscript should be based on the findings and discussion of the manuscript, not by referring to another reference. Therefore, it is suggested that if it is necessary to use the first paragraph of the conclusion section, the corresponding paragraph should be moved to another part of the manuscript (discussion section) and the conclusion section of the manuscript should be based only on the results and discussion conducted in this manuscript.

Thank you for this comment. We have implemented your suggestion. The first paragraph of the conclusion has been removed. Lines 619 to 621 have been reformulated.

• It is necessary to check the entire manuscript for typographical errors.

Thank you for this comment. We have made corrections throughout the manuscript. For example, we have corrected the acronym referring to the questionnaire, removed any extra spaces, and the names of variables are presented in lower case. All of these corrections have been tracked in the revised version of the text.

• It is suggested that the authors check the references and include all the desired fields, including the doi.

Thank you for this comment. We have implemented your suggestion, in particular we have added dois to the references.

Reviewer #1

1- Check an edit as MeSH style.

Thank you for this comment. We have changed some of the keywords as follows: Nursing Homes (D009735), Safety Culture (M000599802), Surveys and Questionnaires (D011795), Safety Management (D017751), Randomised Controlled Trial (D016449) (lines 22 to 23).

2- The abstract is too lengthy. 

Thank you for this comment. We have reviewed the abstract, and we consider that all of the present content is essential for the understanding of the study. We prefer to leave the full names of the dimensions and the statistical results to facilitate their understanding. Furthermore, we have respected the journal's requirement of less than 300 words. However, please do not hesitate to indicate whether there is any particular information you would like us to omit.

3- Mention here about analytical analysis

As suggested, we have modified line 32 to say “Method and Analysis”. If necessary, we can provide details of the statistical analyses that were carried out (Student’s t-test, the type of model used). However, we consider that the summary would then be too detailed.

4- What is the abbreviation?

As suggested, we have explained this abbreviation (line 38). 

5- Pay attention to capital and lower case words.

Thank you for this comment. We have implemented your suggestion. We now present the names of all variables in lower case. These corrections are tracked in the revised version of the text.

6- It is better here to talk about safety culture and climate as general. You can use the following papers to improve the introduction section:

1. A Bayesian network based study on determining the relationship between job stress and safety climate factors in occurrence of accidents

2. The relations of job stress dimensions to safety climate and accidents occurrence among the workers

Thank you for this comment. We have reworked the Introduction and now outline the implementation of safety culture in the healthcare system, along with its importance in risk management. We have quoted the first suggested reference in the Discussion (line 610). 

7- This sentence is vague. Which item?

Thank you for this comment. We have removed the line break in order to improve the flow (lines 100 to 102), and the text now reads as follows: “The assessment of SC helps to raise awareness of patient safety issues, and identify potential areas for improvement. According to Shortell et al., it is one of the key pillars for achieving quality and RM objectives (1).”

8- Reference

As suggested, we have added this information (line 107).

9- ؟؟؟؟

As suggested, the sections of text in question have been reformulated (lines 230 to 232).

10- Pay attention to capital and lower case words

Please see our answer number 5.

11- It is better to show as a figure instead of table

Thank you for this comment. We have replaced Table 4 with a figure (Fig. 2, line 381).

12- In the discussion part, discuss and compare with other studies. 

Thank you for this comment. We have implemented your suggestion. The Discussion has been rewritten. Several bibliographical references have been added.

Reviewer #2

1- In the abstract section line number 30-31 how voluntary participant be inclusion criteria and this may affect the principle of random allocation (justify it)

Our project is an example of population health intervention research (specifically focused on healthcare organisations). This randomised controlled study was aimed at NHs who were members of the SRA. An invitation to participate was first sent by email to all member NHs, and those that volunteered were then contacted by the coordination unit. They were finally included in the study if they met all the inclusion criteria. Therefore, at inclusion, our group of 61 NHs was homogeneous in terms of their willingness to implement the intervention. The criterion of voluntary participation, which was equivalent to the NH’s agreement to take part in the project, was a prerequisite for randomisation. The 61 NHs that had volunteered were then randomly divided into two groups. Consequently, the random allocation of NHs to groups was not affected by our inclusion criteria.

It is possible that whether a NH volunteered to participate in our SC assessment was dependent on their baseline SC level. Baseline scores (T0) have been published (2). However, the paper presented here examines the change in SC scores at T1, which may be attributable to whether or not a support system was implemented in NHs with the same level of commitment at baseline.

2. In the result of abstract section of the line number 41-43, <<After the intervention, the dimension Feedback and communication about incidents significantly improved (+2.8%; p-value=0.044), and so what was dimension of feedback and communication about safety practice or incident before an intervention?

As suggested, we have added this information (line 45). 

3. In line number 53 to 66 in introduction section it is better to write about safety culture implementation in healthcare system and its importance in risk management rather than writing the characteristics of the population in that administrative region.

Thank you for your comment, the Introduction has been reworked to reflect this. We have added lines 56 to 60, and lines 65 to 66. The text describing the characteristics of the population in this administrative region has been substantially shortened, and only lines 69 to 73 have been kept. Five lines have been moved to the Method (lines 146 to 150) as they present variables that are described in the remainder of the manuscript. 

4. Is the tool is validated in your research context?

Thank you for your comment. Yes, the questionnaire has been validated and the validation is reported in another paper. We refer to this in the Summary (line 38) and elsewhere (lines 106, 107, 185 and 283).

5. How determine the sample size for this study purpose?

No formal calculation regarding the required sample size was carried out. In practice, we could not find any previous work (either from institutional structures or in scientific publications) regarding the expected size of the difference in SC scores between control and intervention groups. Technical and financial constraints meant that it was feasible for us to support up to 70 NHs (35 in each group). In the end, 61 were recruited, i.e..15% less than the number initially envisaged.

Our randomised controlled study appears to have no precedent in the literature in terms of the number of organisations participating in the implementation of an in situ support system. However, we did find other interventions that aimed to improve SC. For example, nine NHs participated in a cross-sectional, longitudinal, prospective study (3), and three NHs participated in a non-randomised study with a control and an intervention group (4). Nevertheless, the inclusion of 61 NHs provided sufficient power to show a statistically significant increase for dimension 7 in the intervention group. We believe that if we had included 35 NHs per group, as we had originally envisaged, the power of our analysis would have increased, and we would have been able to demonstrate further significant changes (notably for dimensions 1 and 4).

6. Line number 124, randomization might be affected using voluntary participation of study participants as inclusion criteria rather than other exposure status and please explained it how was the selection process

Thank you for your comment. Randomisation followed inclusion in the project. All NHs (n=61) volunteered to participate. We did not ask NHs about their preferences regarding group membership (control or intervention group).

7. Do you control researcher bias in assignments’ of in intervention in exposed group and unexposed group this is due to the researches should remain unaware of which treatment was given for which group until the study is completed?

Thank you for your comment. We carried out a randomised controlled study of healthcare organisations. It was not a clinical intervention (case-control, single or double-blind). As the NH was aware that an intervention was taking place, there was no reference or placebo. For financial reasons, and other reasons related to the quality and safety of care, and the structure of the support provided, some of the authors of the present manuscript participated in the delivery of the intervention to NHs in the intervention group first, and then in the control group, and the lead author was aware of the distribution of NHs between the two groups.

All data collected in the field were anonymised before statistical analysis and interpretation. Researcher bias was therefore not completely controlled for in the interpretation of the explanatory factors driving the change in SC scores. It is often the case that researcher bias cannot be controlled for in action research studies conducted in the field. Following your remark, we have mentioned this bias in the Strengths and Limitations section (beginning on line 570). In particular, it would indeed be preferable to recruit an external person to interpret the results if the intervention is implemented in future research.

8. Regarding to ethical concern of the research that involve human subject ethical concern is mandatory what types of consent you obtained from the study participants and before or after completion of the questioner participants was give their consent?

Thank you for your comment. Please see lines 308 to 316. We have added the following clarification, “Our research is registered by the Cnil (Commission Nationale de l’Informatique et des Libertés) under ref. 915719 (December 1, 2015)” (line 309).

9. In line number 300-304, the result showed that presence of high non- response rate and did this affect your result?

Thank you for your comment. Although our response rates (64% in 2016, and 56.5% in 2017) are comparable with other studies (66% [10], 69% [29]), their order of magnitude does raise questions about the representativeness of our findings. As it is impossible to determine how this bias impacts the results of our study, we discuss it in the Strengths and Limitations section (beginning on line 567).

10. In your discussion section it is better to start the discussion with the purpose of conducted this research followed by the main finding of your research.

Thank you for this comment. We have implemented your suggestion. The Discussion has been rewritten. Several bibliographical references have been added.

11. The article is needs rewrite in short.

Thank you for this comment. We have reduced the number of words, mainly in the Introduction and the Discussion. The manuscript now has 4976 words, compared to 5981 words in the initial version.

12. The discussion part is shallow and it is better to add different literatures related with the concerned variables and it is so long but it needs short and precise to the concern.

Thank you for this comment. We have implemented your suggestion; the Discussion has been shortened.

13. Limitation and strength is so long it is better write in short.

Thank you for this comment. We have implemented your suggestion, and the word count has been reduced from 459 to 221.

---

## [Decision Letter · Decision Letter 1]

21 Oct 2022

Safety culture in French nursing homes: a randomised controlled study to evaluate the effectiveness of a risk management intervention associated with care

PONE-D-22-13984R1

Dear Dr. Noémie Terrien,

We’re pleased to inform you that your manuscript has been judged scientifically suitable for publication and will be formally accepted for publication once it meets all outstanding technical requirements.

Kind regards,

Jahanpour Alipour, Ph.D.

Academic Editor

PLOS ONE

Reviewers' comments:

Reviewer's Responses to Questions

**Comments to the Author**

1. If the authors have adequately addressed your comments raised in a previous round of review and you feel that this manuscript is now acceptable for publication, you may indicate that here to bypass the “Comments to the Author” section, enter your conflict of interest statement in the “Confidential to Editor” section, and submit your "Accept" recommendation.

Reviewer #1: All comments have been addressed

Reviewer #2: All comments have been addressed

2. Is the manuscript technically sound, and do the data support the conclusions?

Reviewer #1: Yes

Reviewer #2: Yes

3. Has the statistical analysis been performed appropriately and rigorously? 

Reviewer #1: Yes

Reviewer #2: Yes

4. Have the authors made all data underlying the findings in their manuscript fully available?

Reviewer #1: Yes

Reviewer #2: Yes

5. Is the manuscript presented in an intelligible fashion and written in standard English?

Reviewer #1: Yes

Reviewer #2: Yes

6. Review Comments to the Author

Reviewer #1: Dear author,

The paper entitled 'safety culture in French nursing homes: a randomized controlled study to evaluate the

effectiveness of a risk management intervention associated with care' was reviewed again. All comments were addressed. It can be considered for publication. Best

Reviewer #2: Dear editors, thank you give this chance to review this interesting paper and author addressed my all comments, and

the title of abstract is write as <

> and it is unusual in manuscript write up rather in proposal section. Therefore , it is better replaced by << abstract>>

7. PLOS authors have the option to publish the peer review history of their article (what does this mean?). If published, this will include your full peer review and any attached files.

Reviewer #1: No

Reviewer #2: No

---

## [Editor Report · Acceptance letter]

15 Nov 2022

PONE-D-22-13984R1 

Safety culture in French nursing homes: a randomised controlled study to evaluate the effectiveness of a risk management intervention associated with care 

Dear Dr. Terrien:

I'm pleased to inform you that your manuscript has been deemed suitable for publication in PLOS ONE. Congratulations! Your manuscript is now with our production department. 

Kind regards, 

on behalf of

Dr., Jahanpour Alipour 

Academic Editor

PLOS ONE